

# Combined microbiome and metabolomics analysis of yupingfeng san fermented by *Bacillus coagulans*: insights into probiotic and herbal interactions

Yu Kang[1,2], Yanting Sun[1], Jinzhong Cui[2], Yuzhen Song[1], Zilong Sun[2], Huan Li[1], Ruiyan Niu[2] and Hongxing Qiao[1]

[1] College of Veterinary Medicine, Henan University of Animal Husbandry and Economy, Zhengzhou, Henan, China

[2] College of Veterinary Medicine, Shanxi Agricultural University, Jinzhong, Shanxi, China

Corresponding authors
Ruiyan Niu, niuruiyan2000@163.com
Hongxing Qiao, zzmzqhx@163.com

## ABSTRACT

**Background**. Yupingfeng san is a traditional Chinese medicine formula composed of siler, atractylodes, and astragalus. The herbal medicine fermentation process relies on the role of probiotics. *Bacillus coagulans* is a probiotic commonly used to ferment food and drugs. It produces a variety of beneficial metabolites during fermentation. However, the study on the interaction between *B. coagulans* and yupingfeng san is still blank.

**Methods**. During solid-state fermentation of yupingfeng san, we used metabolomics technology and 16S rDNA sequencing to analyze the differential metabolites and microbial flora of *B. coagulans* at 0, 3, 7, 11, and 15 d, which corresponded to groups A0, B3, B7, B11, and B15, respectively. This research explored the correlation between microorganisms and metabolites in fermented compound Chinese medicine.

**Results**. The results revealed a significant difference in species $\beta$ diversity between group A0 and the B groups ($P < 0.01$). At the phylum level, in fermentation groups B3, B7, B11, and B15, the Cyanobacteria relative abundance decreased by 6.69%, 9.09%, 5.74%, and 2.24%, respectively ($P < 0.05$). The Firmicutes relative abundance increased by 39.73%, 35.65%, 49.09%, and 68.66% ($P < 0.05$), respectively. The Proteobacteria relative abundance decreased by 39.86% and 26.70%, respectively, in groups B11 and B15 ($P < 0.05$). The relative abundance of Actinobacteria increased initially with extended fermentation time, and then gradually decreased after reaching its peak in group B7. At the genus level, compared with group A0, the relative abundance of Actinobacteria increased to its highest level of 21.12% in fermentation group B3 and decreased to 9.51% after a fermentation time of 15 d. The abundance of Leuconostoc in fermentation groups B3, B7, and B11 was significantly higher than in group A0 (20.93%, 20.73%, and 21.00%, respectively, $P < 0.05$). Pediococcus in fermentation groups B3, B7, B11, and B15 was also significantly higher than in group A0 (4.20%, 2.35%, 18.84%, and 52.01%, $P < 0.05$). Both Pediococcus and Leuconostoc, which belong to lactic acid bacteria, increased fivefold, accounting for a total abundance of 62%. After yupingfeng san fermentation, using nontargeted metabolomics, we identified 315 differential metabolites. This results showed a decrease in the content of alkene and an increase in the contents of acids, lipids, ketones, and amino acids. In addition, in group B3, the contents of quercetin, paeoniflorin-3-O-glucoside, netin, iristin, anthocyanin, caffeic acid, rosmarinic acid, liquiritin, and isoliquiritin were significantly upregulated.

**Conclusion**. In this study, the composition and metabolic profile of yupingfeng san after the fermentation of *B. coagulans* were studied, and it was found that the fermentation group showed rich species diversity, in which the abundance of Leuconostoc and Weisseria increased significantly, while the opportunistic pathogens such as *Pseudomonas aeruginosa* and Enterobacter decreased significantly. The analysis of metabolic products showed that the contents of acids, lipids and ketones were significantly increased, rich in a variety of beneficial microorganisms and small molecular compounds with antibacterial effects, and these changes worked together to inhibit the growth of pathogens and maintain intestinal health. The study not only helps to elucidate the assembly mechanism and functional expression of microorganisms after Chinese traditional medicine fermentation, but also provides a solid scientific basis for the development of efficient and safe micro-ecological feed additives.

## INTRODUCTION

Traditional Chinese medicine (TCM) is applied to both treat and prevent several diseases. Thanks to its low toxicity and limited side effects, TCM has become a popular research topic. Zhu Danxi of the Yuan Dynasty in China first wrote about yupingfeng san in "Dan Xi Xin Fa". Yupingfeng san is based on the use of *astragalus* as the main medicine and is supported by *atractylodes rhizomes* and *fangfeng*. This herbal formula has been traditionally prescribed to enhance the body's defense mechanisms against pathogens and is particularly noted for its application in the prevention and treatment of respiratory infections, including those caused by viruses. Recent studies have explored the efficacy and safety of yupingfeng powder, revealing its potential benefits in managing allergic rhinitis and other chronic inflammatory conditions (*Cheong et al., 2022*; *Xiong & Qian, 2013*). Yupingfeng san can effectively treat respiratory diseases and control respiratory inflammation and allergic reactions. Some studies have shown that yupingfeng san polysaccharides have a dual immunomodulatory effect of regulating the proliferation and cytokine expression of lymphocytes and macrophages in mice (*Sun et al., 2017*). By limiting the proliferation of intestinal enterochromaffin cells and 5-hydroxytryptamine content, yupingfeng san had an anti-inflammatory effect in trinitrobenzene sulfonic acid-induced colitis in mice. It has been speculated that the excellent immunomodulatory function of yupingfeng san confers it with great potential for clinical application (*Zang et al., 2015*). According to several studies, yupingfeng san has been shown to have good anti-immune and anti-inflammatory activities and has had a therapeutic and preventive effect on COVID-19. Most studies have examined the molecular biological and pharmacological verification of yupingfeng san. To date, however, its mechanism of action from the level of metabolites and 16S rDNA has not been fully examined.

In China, the use of fermentation to change the efficacy of Chinese herbs has a long history. Through evolution and development, with the continuous addition of modern technology, fermentation has gradually developed into a common and important traditional Chinese medicine processing method. Fermenting TCM with microorganisms according to the use of appropriate humidity, temperature, and water conditions can improve the original efficacy. As a result, it is possible to reduce the dosage of drugs as well as expand their application. The two types of TCM fermentation technology are solid-state and liquid fermentation. Solid-state fermentation, which is characterized by fermenting solid substrates according to low-moisture conditions, has a long history in China. Because of the low water content in the fermentation process, it has few site requirements as well as low cost requirements. This method is more suitable for industrial promotion, and the product state is also more suitable for clinical application. In liquid fermentation (also called liquid immersion fermentation), the strain is activated and cultured to a good growth state and then is inoculated into the culture medium (composed of medium and Chinese herbal medicine fully mixed at a certain proportion). Fermentation occurs at a suitable temperature after stirring completely (*Li et al., 2020*). *Bacillus coagulans* have shown remarkable results in enhancing the pharmacological properties of Chinese herbs. This probiotic microorganism has been recognized for its potential to improve the efficacy of various herbal formulations, particularly in the context of traditional Chinese medicine (TCM). For instance, studies have demonstrated that *B. coagulans* can significantly influence the bioavailability and therapeutic effects of herbal extracts, leading to enhanced anti-inflammatory and anticancer activities (*Dolati et al., 2021*). Moreover, the incorporation of *B. coagulans* into herbal preparations has been linked to improved immune responses. Research indicates that the polysaccharide adjuvants derived from Chinese medicinal herbs, when combined with *B. coagulans*, can stimulate robust immune reactions, thereby augmenting the overall effectiveness of vaccines and herbal treatments (*Wang et al., 2021*). This synergistic effect is particularly noteworthy in the context of chronic diseases, where the modulation of immune pathways is crucial for therapeutic success.

Probiotics are essential in maintaining intestinal function and treating intestinal flora imbalance. Lactic acid bacteria, *Lactobacillus plantarum*, and *Enterococcus faecalis* are commonly used probiotics in TCM fermentation. Previously, we observed significant effects on serum biochemical indices, growth performance, and fecal microbiota of broilers after dietary supplementation of 0.5% astragalus fermented by *L. plantarum* (*Qiao et al., 2018*). Studies have shown that optimizing *Lactobacillus pentosus* Stm fermentation of Astragalus can improve the performance and egg quality of laying hens by improving the intestinal microbiota (*Dong et al., 2023*). However, with the use of a single Chinese herb astragalus in these study, the biological activity and efficacy of its fermentation products may be relatively single, which can not give full play to the synergistic effect of compound fermentation. In addition, the study mainly focused on the effects of fermentation on poultry growth performance and microbial community, and did not explore the changes of metabolites in depth. Because the genera *Lactobacillus* and *Bifidobacterium* have a nonspore nature, they must be freeze-dried. As a result, the shelf life of powdered probiotic

preparations is reduced and their viability is adversely affected (*Govender et al., 2014*). Previous studies on the relationship between probiotics and traditional Chinese medicine mainly focused on a few probiotics such as *Lactobacillus* and *Bifidobacterium*, which limited the microbial diversity in the fermentation process and ignored other potentially beneficial microorganisms such as *B. coagulans*. In addition, with the development of the Chinese herbal medicine fermentation industry and related research, there is an urgent need for probiotics for fermentation with more functions. In this regard, *Bacillus coagulans* holds significant research value. In addition to its considerable acid-producing capacity, *B. coagulans* can prevent intestinal diseases, and its ability to inhibit harmful bacteria is stronger than that of traditional lactic acid bacteria or *Enterococcus faecalis* (*Hyronimus, Le Marrec & Urdaci, 1998*). Additionally, *B. coagulans* has been shown to interact beneficially with the gut microbiome, which plays a vital role in the metabolism of herbal compounds. By promoting a healthy gut environment, *B. coagulans* can enhance the absorption and efficacy of bioactive constituents found in Chinese herbs, such as flavonoids and saponins, which are known for their health-promoting properties (*Abdallah et al., 2019*). Furthermore, *B. coagulans* can form spores during growth, which makes it highly resistant to acid, and it can maintain its activity in a sealed state for a long time. Related products have a longer shelf life, and it is easier for probiotics to reach the gut after use. After the spores of *B. coagulans* reach the duodenum, they become nutrients to play their probiotic role, and then are excreted from the body, which is why they have little effect on the intestinal microbial community. In terms of safety, *B. coagulans* is a "generally considered safe". *Bacillus* approved by the US Food and Drug Administration. It has been used in the field of dietary supplements and medicine and has achieved good efficacy. It was found that fermentation of *Ginkgo biloba* by *B. coagulans* promoted the beneficial effects of *G. biloba* on the immune status of broilers (*Liu et al., 2015*), and it also had significant effects on the abdominal fat deposition and meat quality of Peking duck (*Liu et al., 2017*). Furthermore, because of its excellent immunomodulatory properties, antiviral activity, and ability to regulate intestinal microbial balance, *B. coagulans* has significant potential in the field of feed additive.

## MATERIALS AND METHODS

### Preparation of microbial and fermented yupingfeng san

We used *B. coagulans* (NJ001, isolated and preserved in the Henan Province Engineering Laboratory of Microbial Biotransformation of Traditional Chinese Medicine) to ferment yupingfeng san. We grew *B. coagulans* in Man Rogosa Sharpe (MRS) medium. It was incubated at 37 °C for 48 h. Before fermentation, we stored yupingfeng san in a cool and ventilated place for later use. The activated *Bacillus* condensing liquid was evenly mixed with yupingfeng san (40-mesh screen) at a proportion of 35% of the total mass (the ratio of yupingfeng san and the liquid phase was 6:4, with additional water to reach a specified volume-the humidity in the fermentation system was 40%). After mixing, we divided the mixture into 35 mm × 45 mm plastic film fermentation bags with vent valves, and fermented it at a constant temperature incubator of 37 °C for 0–15 d. At the beginning of

fermentation (0 d), the pH was between 5.0 and 5.5, and the pH was lower than 5.0 from 5 days to the end of fermentation. We used an ultraclean table for multipoint sampling after fermentation for 0, 3, 5, 7, and 15 d. We collected and stored the samples at $-80\ °C$ to subsequently determine metabolites and bacterial composition.

## DNA extraction and data analysis

DNA from 21 TCM samples (six replicates in group $A_0$, six replicates in group $B_3$, three replicates in group $B_7$, three replicates in group $B_{11}$, and three replicates in group $B_{15}$) was extracted using the cetyltrimethylammonium bromide (CTAB) according to manufacturer's instructions by QIAGEN DNeasy PowerSoil Pro Kit. The reagent which was designed to uncover DNA from trace amounts of sample has been shown to be effective for the preparation of DNA of most bacteria. Nuclear-free water was used for blank. The total DNA was eluted in 50 μL of Elution buffer and storedat $-80\ °C$ until measurement in the PCR by LC-Bio TechnologyCo., Ltd, Hang Zhou, Zhejiang Province, China.

The 5′ends of the primers were tagged with specific barcods per sample and sequencing (*Logue et al., 2016*). Validation of the PCR amplification was conducted utilizing a 2% agarose gel electrophoresis technique. In the DNA isolation phase, ultrapure water served as a negative control, replacing the standard sample solution to mitigate the risk of spurious positive PCR amplifications. Post-amplification, the PCR products underwent purification *via* AMPure XT beads procured from Beckman Coulter Genomics (Danvers, MA, USA). Quantitation of these purified products was facilitated by the Qubit system, a product of Invitrogen (Waltham, MA, USA).

Subsequent to purification, amplicon pools were assembled in preparation for sequencing endeavors. The integrity and abundance of the amplicon libraries were gauged using the Agilent 2100 Bioanalyzer, complemented by the Library Quantification Kit for Illumina, the latter being sourced from Kapa Biosciences (Woburn, MA, USA). The sequencing of these libraries was executed on the NovaSeq PE250 platform.

Adhering to the protocols stipulated by the manufacturer and as advised by LC-Bio, the samples were sequenced on the Illumina NovaSeq platform. The resultant paired-end reads were attributed to their corresponding samples leveraging unique barcodes, with the barcode and primer sequences being excised to truncate the reads. These truncated reads were subsequently amalgamated employing the FLASH algorithm (*Magoč & Salzberg, 2011*). Quality control of the raw sequencing data was executed under stringent filtering criteria, yielding high-quality clean tags as per the fqtrim software (version 0.94). Chimeric sequences were discerned and eliminated using the Vsearch software (version 2.3.4) (*Rognes et al., 2016*). Following the dereplication process with DADA2 (*Callahan et al., 2016*), a feature table and associated sequences were derived. Alpha and beta diversity analyses were conducted after normalizing the sequence data to a uniform count through random subsampling. The SILVA database (release 138) (*Quast et al., 2013*) was employed to normalize the feature abundance based on the relative abundance per sample. Alpha diversity analysis, which assesses the richness and evenness of species within a sample, was performed using five indices: Chao1, Observed species, Goods coverage, Shannon, and Simpson. The computation of these indices was facilitated by the QIIME2 platform. Beta

diversity analysis, which evaluates the dissimilarity between samples, was also computed using QIIME2, with graphical representations being generated to visualize the results. Data analysis and 16S rDNA gene sequencing were performed by LC-Bio TechnologyCo., Ltd, (Hangzhou, Zhejiang Province, China).

## Metabolite extraction, liquid chromatography/mass spectrometry, and data processing

We collected a total of 12 samples of traditional Chinese medicine (six replicates in group $A_0$ and six replicates in group $B_3$), when the concentration of bacterial liquid reached a peak after *B. coagulans* fermentation of yupingfeng san for 3 d after a preliminary test. We weighed 100 mg of each sample, which we ground in liquid nitrogen and then placed in a 1.5-mL Eppendorf tube. We added 500 mL of a 50% methanol/water solution by sonicating the mixture. We stored the extraction mixture in a refrigerator for 2 h at $-20\,°C$, which was centrifuged for 10 min at 20,000 g. After removing the supernatant with a pipettor, we drained it with a cryo-extractor. We redissolved 100 $\mu L$ of acetonitrile and removed 10 $\mu L$ of the dilution to mix the QC samples. We stored the samples at $-80\,°C$ in a refrigerator until we conducted the mass spectrometry analysis.

We used the LC-MS system following the manufacturer's equipment instructions to collect all samples. We used a Vanquish Flex UHPLC system (Thermo Fisher Scientific, Waltham, MA, USA) to perform all chromatographic separations. We used an ACQUITY UPLC T3 column (100 mm × 2.1 mm, 1.8 $\mu m$, Waters, Milford, CT, USA) to perform reversed-phase separation and maintained the column oven at 35 °C and at a flow rate of 0.4 mL/min. The mobile phases included solvent A (water, 0.1% formic acid) and solvent B (acetonitrile, 0.1% formic acid). We set the gradient elution conditions as follows: 0–0.5 min, 5% B; 0.5–7 min, 5–100% B; 7–8 min, 100% B; 8–8.1 min, 5–100% B; and 8.1–10 min, 5% B. We used a high-resolution tandem mass spectrometer Q-Exactive (Thermo Fisher Scientific, Waltham, MA, USA) for detection to elute the metabolites from the column. We performed Q-Exactive analyses in the negative and positive ion modes. To achieve an automatic gain control target of 3e6, we collected precursor spectra (70–1,050 m/z) at a resolution of 70,000 and set the maximum injection time to 100 ms. To acquire data for the digital differential analyzer mode for the top three configurations, we used the top three configurations. To reach the automatic gain control target and evaluate the stability of samples determined by LC-MS with a maximum injection time of 80 ms, we collected fragment spectra at a 17,500 resolution. Throughout the collection period, we collected quality control samples (a pool of all samples) every 10 samples.

We used Proteowizard MSConvert software to convert the mass spectrometry raw data into readable mzXML data. We used XCMS software for quality control of peak extraction and peak extraction. We used CAMERA and MetaX software to add, annotate, and then identify the extracted substances *(Wen et al., 2017)*. To identify secondary mass spectrum information, we match the primary mass spectrum information to an in-house standard database. We used HMDB, KEGG, and other databases to annotate the candidate-identified substances and to explain the biological functions and physicochemical properties of metabolites. To quantify and screen the differential metabolites we used MetaX software.

Hangzhou Lianchuan Biotechnology Co., Ltd. performed data processing and liquid chromatography/mass spectrometry analysis.

## Data analysis

We used SPSS Statistics software version 20.0 (IBM Corp., Armonk, NY, USA) to perform statistical analyses. To evaluate significant differences in microbial genera and phyla between different groups, we used the Kruskal–Wallis test, and to evaluate significant differences between different groups, we used the $t$-test and $P < 0.05$. We used the false discovery rate (FDR) to correct $P$ values for metabolomics and microbial data. We considered FDR-corrected $P$ values below 0.05 to be statistically significant. We used SIMCA-P software to perform partial least squares-discriminant analysis (PLS-DA). According to goodness-of-fit parameter (R2X) and the predictive ability parameter (Q2), we determined PLS-DA, and based on these results, we drew metabolite maps according to their importance. We assigned a value-variable importance in projection (VIP) to each variable, and VIP greater than 1.0 indicated significant differences. To assess metabolites and microbiota using GraphPad Prism version 6.00 (GraphPad Software, San Diego, CA, USA), we calculated Spearman rank correlation coefficients.

# RESULTS

## $\alpha$ diversity analysis of yupingfeng san fermented by *Bacillus coagulans*

A total of 1,752,289 original sequences and 1,583,897 valid sequence reads were obtained from the five groups of samples, and the average number of valid sequence reads per sample was 75,423. Table 1 and Fig. 1 show the $\alpha$ diversity of the microbial community of yupingfeng san fermented by *B. coagulans*. We detected the group $B_3$ had the highest Shannon index, and group $B_7$ had the second highest, which indicated the highest diversity and abundance of the microbiota in the samples after fermentation for 3 d. The Simpson and Chao1 indices and observed operational taxonomic units (OTUs) of the unfermented groups were higher than those of the fermented group $A_0$. These values decreased as fermentation time increased, which showed that the richness and diversity of microbial colonies in yupingfeng san could be reduced after fermentation. As shown in Fig. 1, the Shannon index of samples in group $B_3$ (6.2567 $\pm$ 0.20207) was higher than that in group $A_0$ (6.04 $\pm$ 0.24759), but the difference was not significant ($P \geq 0.05$), indicating that the microbial species in group $B_3$ was slightly higher than that in group $A_0$. The Shannon index of samples in $B_3$ group (6.2567 $\pm$ 0.20207) was higher than that in $B_{11}$ group (5.4933 $\pm$ 0.36460) and $B_{15}$ group (4.2367 $\pm$ 0.34530), and the difference was significant ($P \leq 0.05$), indicating that the microbial community diversity in $B_3$ group was higher than that in $B_{11}$ and $B_{15}$ groups. The Shannon index of $A_0$ group (6.04 $\pm$ 0.24759) was higher than that of $B_{11}$ group (5.4933 $\pm$ 0.36460) and $B_{15}$ group (4.2367 $\pm$ 0.34530), and the difference was significant ($P \leq 0.05$).

**Table 1** $\alpha$ Diversity measurement of the bacterial community in yupingfeng san samples fermented by *Bacillus coagulans*.

| Group | Observed OTUs | Shannon | Simpson | Chao1 | Pielou's |
|---|---|---|---|---|---|
| $A_0$ | 558.00 | $6.04 \pm 0.24759^c$ | $0.956 \pm 0.00577^b$ | $559 \pm 94.43926^c$ | 0.68 |
| $B_3$ | 435.33 | $6.2567 \pm 0.20207^c$ | $0.953 \pm 0.00577^b$ | $429.95 \pm 12.51476^c$ | 0.71 |
| $B_7$ | 329.33 | $6.1167 \pm 0.15503^c$ | $0.956 \pm 0.01155^b$ | $329.916 \pm 106.95511^{ab}$ | 0.74 |
| $B_{11}$ | 304.00 | $5.4933 \pm 0.36460^b$ | $0.923 \pm 0.03055^b$ | $305.7333 \pm 29.11818^{ab}$ | 0.67 |
| $B_{15}$ | 254.00 | $4.2367 \pm 0.34530^a$ | $0.79 \pm 0.05568^a$ | $255.5667 \pm 22.30106^a$ | 0.53 |

**Notes.**

Statistically significant differences ($P \leq 0.05$) are denoted by different lowercase letters in the same column. Differences that are not significant ($P \geq 0.05$) are denoted by the same lowercase letters or no letters in the same column.

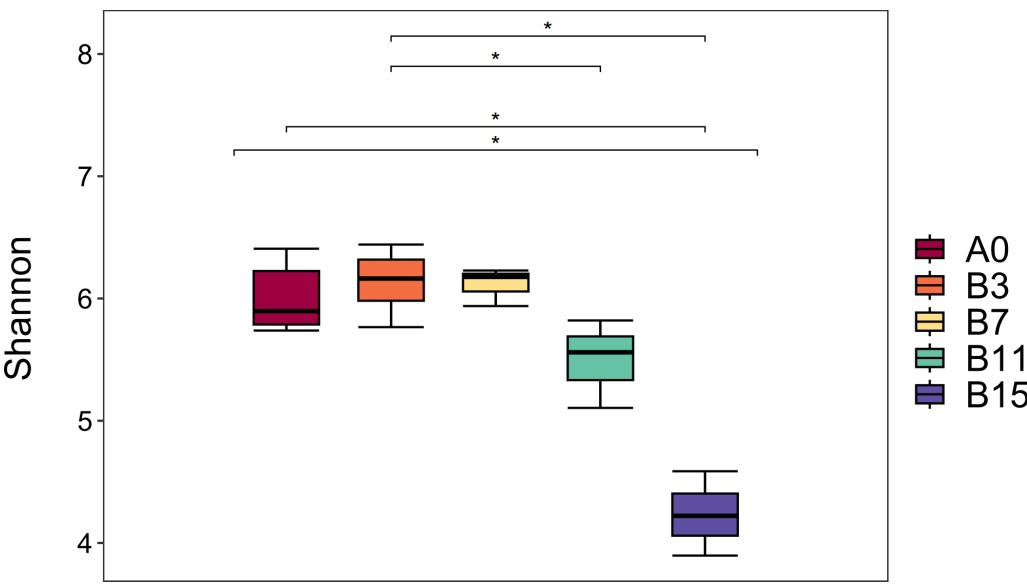

**Figure 1** Boxplot diagram of Alpha diversity analysis of yupingfeng powder samples fermented by *Bacillus coagulans*.

## $\beta$ diversity analysis of yupingfeng san fermented by *Bacillus coagulans*

The principal coordinate analysis (PCoA) plot shows the changes in the microbial community composition of the 21 samples according to the $\beta$ diversity analysis. The unfermented and fermented groups (Fig. 2) had a clear separation of the microbial community, which showed that the microbial community composition changed after the fermentation treatment with *B. coagulans*. PCoA1 accounted for 32.45% of the total variation, and PCoA2 accounted for 20.33%. The species in group $A_0$ were significantly different from those in group B, but the microbial community species of the samples were similar. The microbial community species of samples in groups $B_3$, $B_7$, and $B_{11}$ were similar and overlapping. The microbial community of samples in the $B_{15}$ group was significantly

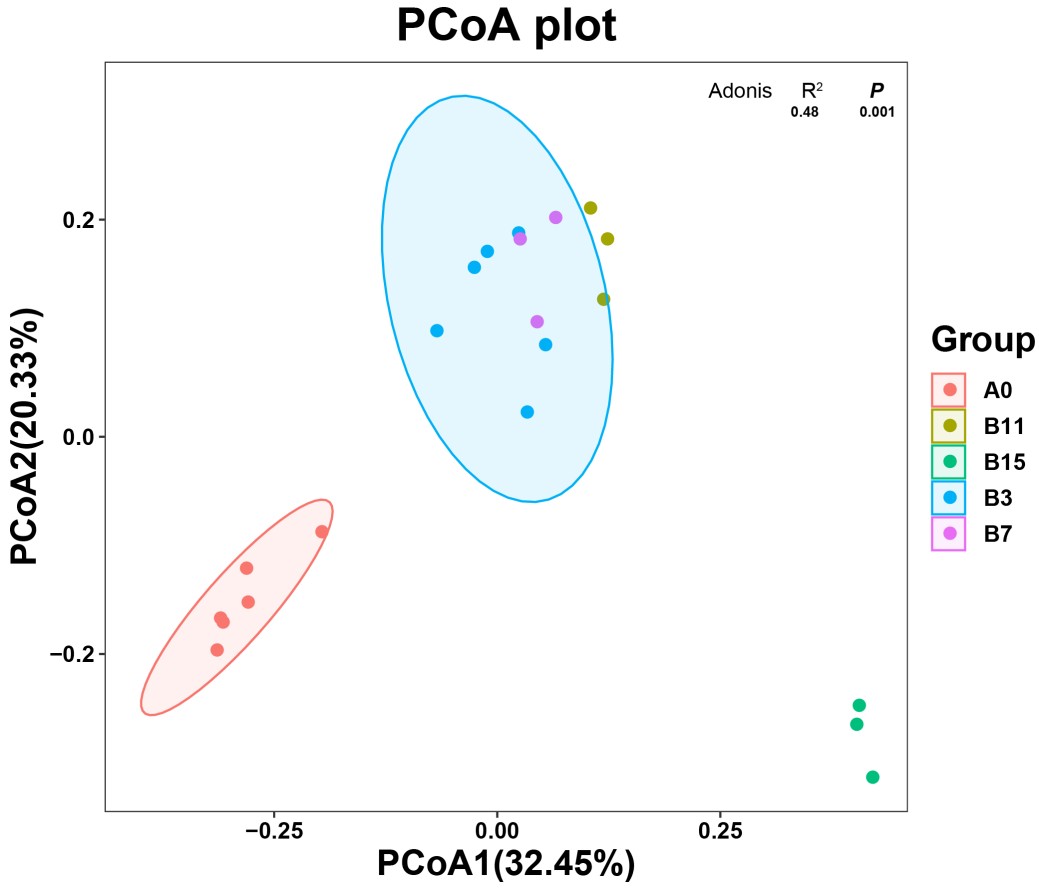

**Figure 2** Species diversity of yupingfeng san fermented by *Bacillus coagulans*.

different from the other fermentation groups and these samples were part of the same cluster.

## Composition of the microbial flora of yupingfeng san fermented by *Bacillus coagulans*

At the phylum level, according to the species abundance table and the species annotation table, we identified the top 26 phylum categories (Fig. 3A). *Proteobacteria* (45.21%), *Firmicutes* (32.50%), *Cyanobacteria* (19.87%), *Bacteroidetes* (0.99%), and *Actinobacteria* (0.93%) were the most abundant species in group $A_0$. The relative abundance of *Firmicutes*, *Proteobacteria*, and *Cyanobacteria* changed with an extended fermentation time in group B. In groups $B_3$, $B_7$, $B_{11}$, and $B_{15}$ the relative abundance of *Firmicutes* significantly increased (39.73%, 35.65%, 49.09%, and 68.66%, respectively, $P < 0.05$) and the relative abundance of *Cyanobacteria* significantly decreased (6.69%, 9.09%, 5.74%, and 2.24%, respectively, $P < 0.05$). In groups $B_{11}$ and $B_{15}$, the relative abundance of *Proteobacteria* significantly decreased (39.86% and 26.70%, respectively, $P < 0.05$). In group $B_7$, the relative abundance of *Actinobacteria* first increased with the extended fermentation time, and after reaching the peak, gradually decreased.

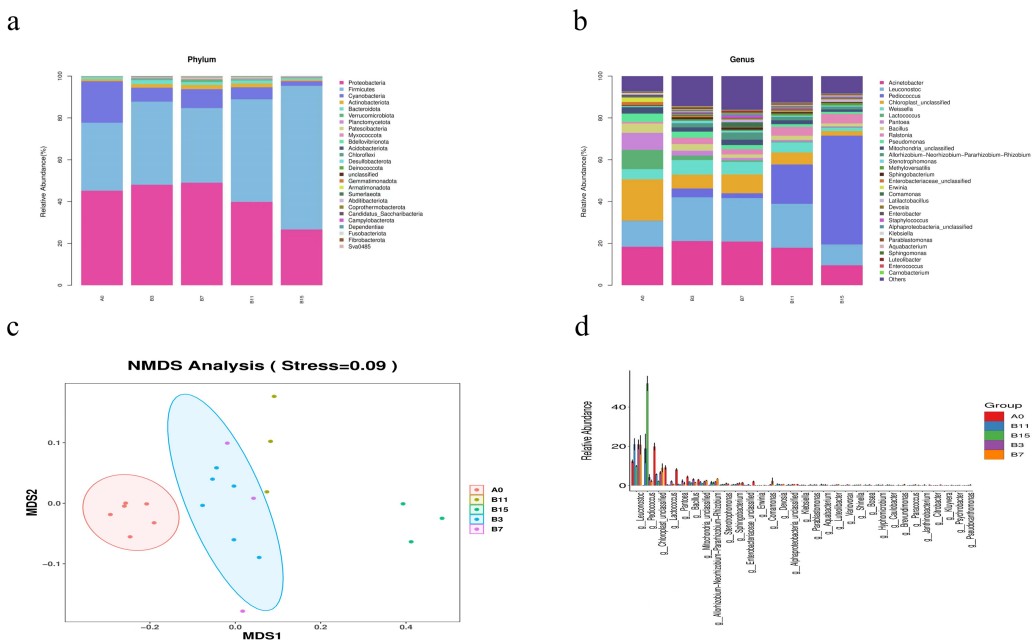

**Figure 3** **Genus- and phylum-level analysis of 21 samples of yupingfeng san fermented by *Bacillus coagulans*.** (A) Microbiota composition of group $A_0$ and each B group sample at the phylum level. (B) Microbiota composition at the genus level of samples from $A_0$ and each B group. (C) NMDS analysis diagram based on ASV abundance of each sample. (D) NMDS analysis diagram based on ASV abundance of each sample.

At the genus level, the sequences of the 21 samples had 32 major genera (Fig. 3B). In group $A_0$, the most common genera were *Acinetobacter* (18.37%), *Leuconostoc* (12.33%), *Lactococcus* (9.20%), *Pantoea* (8.15%), *Weissella* (4.85%), and *Bacillus* (4.47%). Compared with the relative abundance of group $A_0$, that of *Acinetobacter* increased to its highest value of 21.12% in fermentation group $B_3$, and decreased to 9.51% when the fermentation time was extended to 15 d. *Leuconostoc* in fermentation groups $B_3$, $B_7$, and $B_{11}$ was significantly higher than in group $A_0$ (20.93%, 20.73%, and 21.00%, respectively, $P < 0.05$). *Pediococcus* in fermentation groups $B_3$, $B_7$, $B_{11}$, and $B_{15}$ was significantly higher than in group $A_0$ (4.20%, 2.35%, 18.84%, and 52.01%, respectively, $P < 0.05$), and *Pediococcus* and *Leuconostoc*, which belonged to lactic acid bacteria, increased fivefold, accounting for 62.0% of the abundance. In group $A_0$, the number of Rohlstone-producing bacteria was significantly lower than in group $B_3$ (4.58%). The relative abundance of *Weissella* increased to its peak (6.97%) in group $B_3$, and then decreased after fermentation for 7–15 d. With prolonged fermentation time, the relative abundance of *Pseudomonas* gradually decreased.

The effect of *B. coagulans* fermentation on the bacterial colony of yupingfeng san was obvious. NMDS model showed that there were obvious differences between group B samples fermented by *B. coagulans* and unfermented group A samples of yupingfeng san (Fig. 3C). The difference comparison of dominant bacteria genera showed that *Leuconostoc* and *Pediococcus* were significantly more abundant in fermented yupingfeng san than in the non-fermented group, while *Lactococcus* and *Pantoea* were significantly less abundant

in fermented yupingfeng san than in the non-fermented group. The abundance of the important pathogenic bacterium *Pseudomonas* and the potential pathogen *Enterobacter* decreased significantly in the fermentation group, indicating that fermentation is beneficial to animal health (Fig. 3D).

## Differential metabolites of yupingfeng san pre- and post-fermentation

PLS-DA is a supervised differential discriminant analysis method, which can reflect the difference between different groups to the greatest extent. As shown in Fig. 4A, samples from different groups are clustered into distinct time intervals. Repeated sample aggregation in the same group indicates a high degree of similarity in the observed variables. In general, R2 represents the model interpretation rate and Q2 represents the model prediction rate, and the two values are better than 0.5, and the closer to 1, the better. As shown in Fig. 4B, R2 and Q2 in the replacement test model were both greater than 0.5, R2 was 0.9769 close to 1, and Q2 was $-0.6349$ less than 0, indicating that there was no overfitting in the model and the differential metabolite analysis was more accurate (Fig. 4B). We used a combination of the $P$ value of $t$-tests, their ratio, and VIP value of the PLS-DA model to screen the differential metabolites between groups $A_0$ and $B_3$. We used the following conditions: $P < 0.05$, ratio $\geq 2$ or ratio $\leq 1/2$, and VIP $\geq 1$. To show significant differences between the fermented and the unfermented Chinese medicine group, we screened 315 metabolites. Figure 4C shows a volcano diagram for the differential metabolites. We found that 200 metabolites were significantly upregulated and 115 metabolites were significantly downregulated in the fermented Chinese medicine group. In group $B_3$, the contents of quercetin, paeoniflorin-3-O-glucoside, netin, iristin, anthocyanin, caffeic acid, rosmarinic acid, liquiritin, and isoliquiritin were significantly upregulated.

## Metabolic pathway analysis

According to $P$-value , the key metabolic pathways were identified by KEGG pathway enrichment analysis. Figure 5 shows the enrichment factor, which represents the number of differential metabolites per number of metabolites in the KEGG analysis. The smaller the $P$ value, the degree of KEGG enrichment was higher. According to the pathway enrichment analysis of differential metabolites, we obtained the following metabolic pathways: 127 pathways were enriched in metabolic phases, including energy, lipid, and amino acid (*e.g.*, alanine, aspartate, glutamate metabolism, phenylalanine metabolism, and tyrosine metabolism; $P < 0.05$). We also found that 85 metabolic pathways were enriched in the following secondary metabolites (*e.g.*, monoamine biosynthesis and novobiocin biosynthesis; $P < 0.05$); 58 microbial metabolic pathways were enriched in different environments; and 25 amino acid biosynthesis pathways were enriched.

## Microbial flora and metabolites in yupingfeng san fermented by *Bacillus coagulans*

We drew a related heat map using the OTUs and metabolites of yupingfeng san fermented by *B. coagulan*s to identify the internal relationship between the micorobial community and metabolites of yupingfeng san fermented by *B. coagulans*. Figure 6 shows that *Pediococcus*

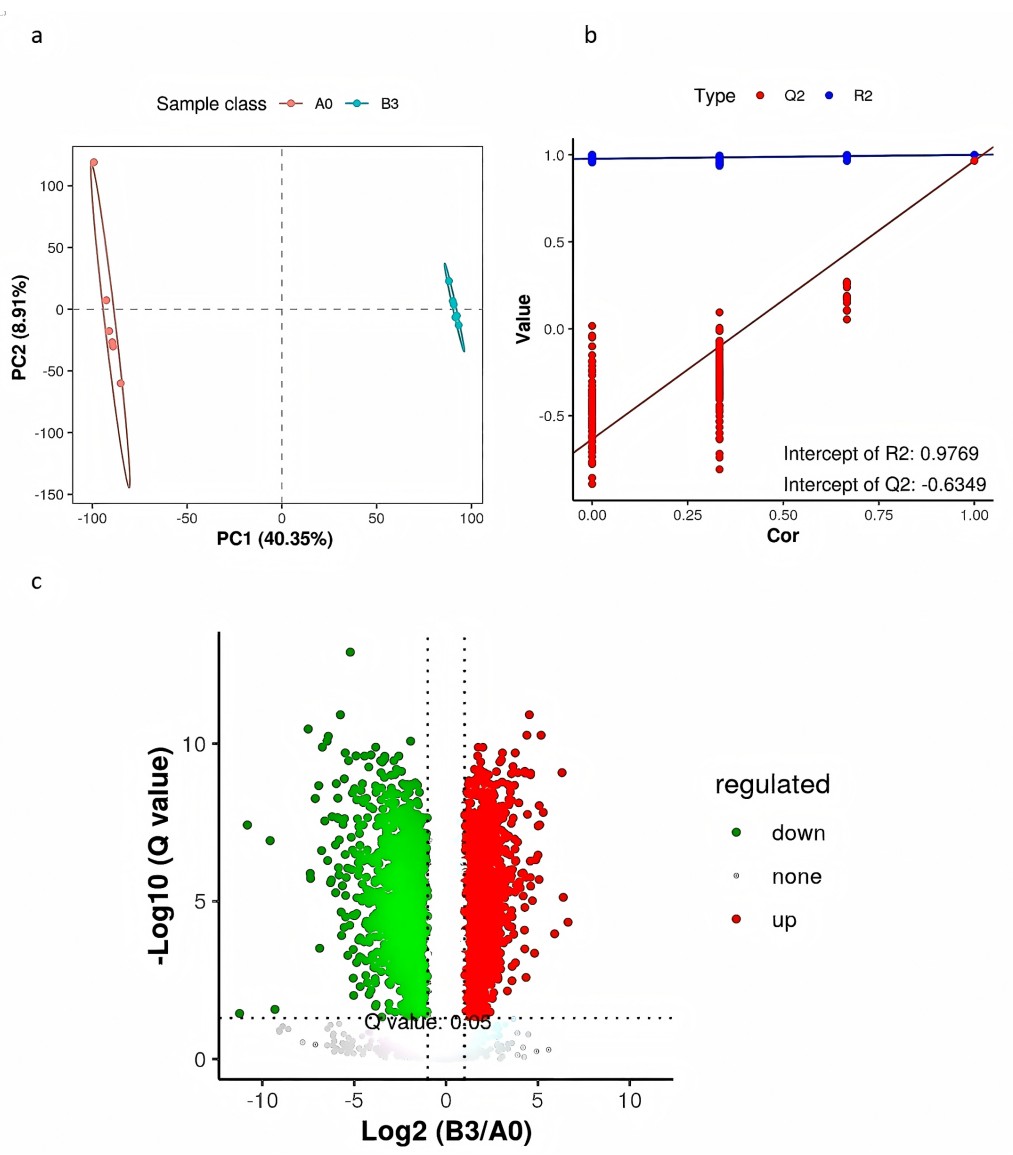

**Figure 4  PLS-DA analysis and differential metabolite volcano map of yupingfeng san pre- and post-fermentation.** (A) PLS-DA score of yupingfeng san pre- and post-fermentation. (B) Permutation test diagram of yupingfeng san pre- and post-fermentation. (C) Volcano map of differential metabolites.

was positively correlated with quercetin but negatively correlated with pelargonidin-3-O-glucoside, and 3′, 4′-dimethoxy-3-hydroxy-6-methylflavone was positively correlated with *Pseudescherichia*.

Chloroplast_unclassified was negatively correlated with the Arg-Leu, and *Parablastomonas* was positively correlated with the flavone Ile-Gly. The genus *Bosea* was negatively correlated with lysophospholipid 18:2. *Aquabacterium* and phlorin were positively correlated. *Caulobacter* and D-(+)-trehalose were negatively correlated. *Hyphomicrobium* and gentisic acid were positively correlated. *Achromobacter* and the branched fatty acid

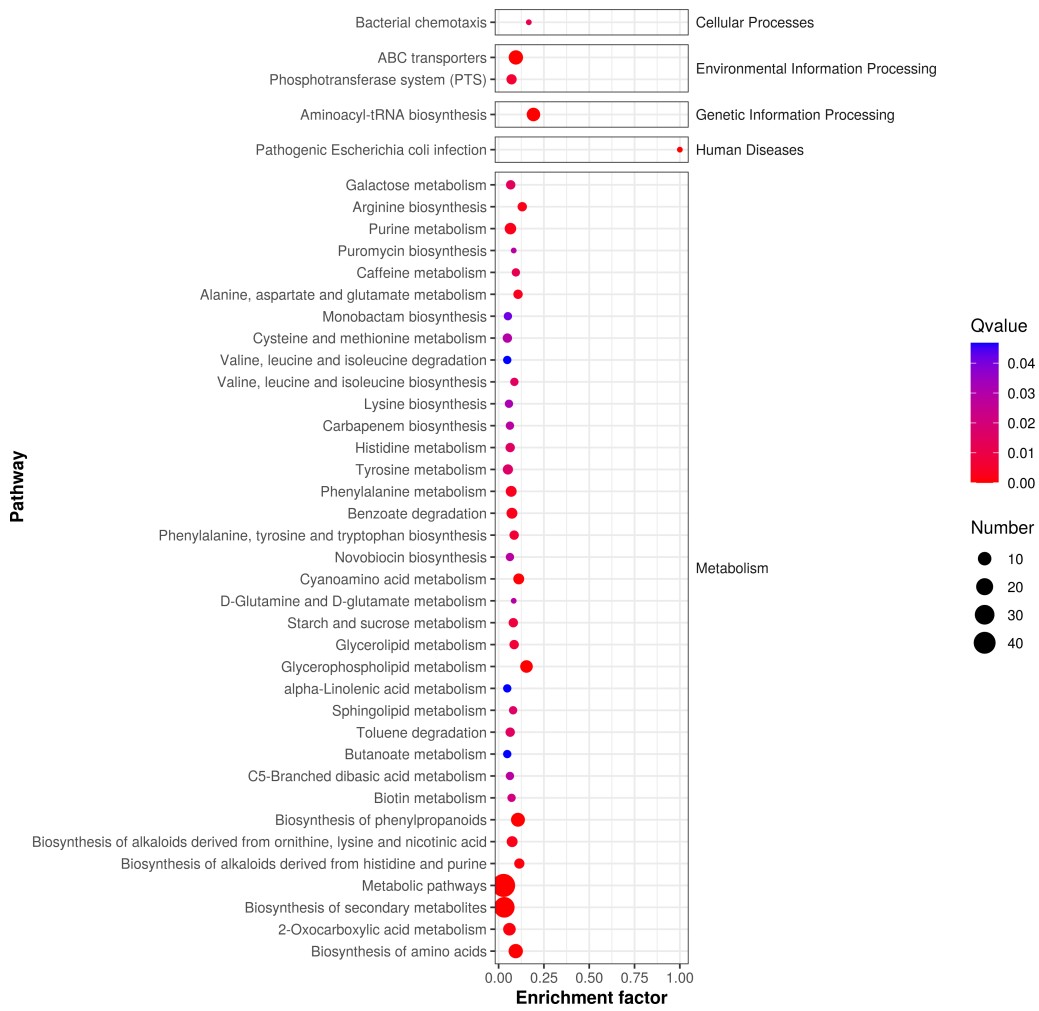

**Figure 5** Enrichment analysis of differential metabolic pathways in fermented yupingfeng san.

esters of hydroxy fatty acid 20:2 were negatively correlated, but *Achromobacter* and 5,7,2′-trihydroxyflavone and 5,7,3′-trihydroxy-4′-methoxyflavanone were positively correlated. *Staphylococcus* and lysophosphatidyl ethanolamine 18:3 were negatively correlated.

## DISCUSSION

The addition of *B. coagulans* had a significant effect on the final microbial population of the fermented Chinese medicine yupingfeng san. Different microbial populations formed in groups $A_0$ and B before and after fermentation. Group $A_0$ has fewer beneficial microorganisms and small molecule compounds with high antimicrobial activity than the B groups. Group $A_0$, however, had more potential pathogens. The number of *Proteobacteria* in fermented yupingfeng san significantly decreased compared with unfermented yupingfeng san. *Proteobacteria* is an important phylum of pathogenic bacteria. Through fermentation, the number of *Proteobacteria* decreased, which directly and effectively improved the safety

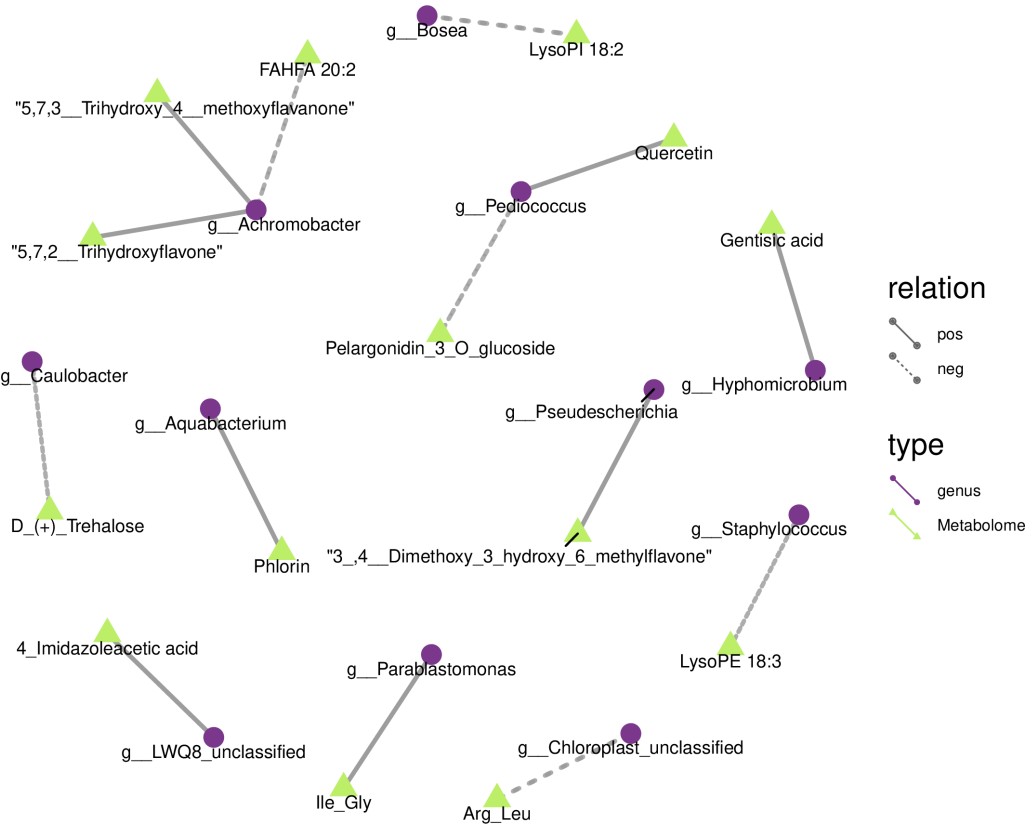

**Figure 6  Metabolites and 16S network regulation analysis results.**

of fermented yupingfeng san. At the same time, the number of *Firmicutes* increased dramatically. The increase in *Firmicutes* is particularly noteworthy, as this phylum has been associated with various health benefits, including the enhancement of gut health and the modulation of immune responses (*Hui et al., 2020a*; *Hui et al., 2020b*). Because *Firmicutes* is one of the main producers of short-chain fatty acids in the intestine, especially butyric acid, a significant increase can improve short-chain fatty acid production and the intestine's immune system function *(Eeckhaut et al., 2011)*. The known important probiotics *Leuconostoc* and *Weissella* were less abundant in unfermented yupingfeng san than in fermented yupingfeng san, which are the core strains of lactic acid bacteria and obligate heterofermentative lactic acid bacteria. Studies have shown that specific strains of *Lactococcus*, such as Lactococcus lactis, can play a crucial role in fermentation processes, contributing to the production of bioactive compounds and improving the overall quality of fermented foods *(Huang et al., 2018)*.

The relative abundance of *Leuconostoc* and *Pediococcus* in fermented yupingfeng san was significantly higher than that in the unfermented group, and this phenomenon was also observed in other fermented products. For example, *Leuconostoc* and *Pediococcus* are considered to be the main *lactobacillus* genera in fermented alfalfa silage, especially when fructose and pectin are added, the abundance of these bacteria genera increases significantly,

thus improving the quality of the silage (*Wang et al., 2020a*; *Wang et al., 2020b*). *Pediococcus* is known for its production of antimicrobial peptides, such as Pediocin, which inhibit the growth of other competing microorganisms, giving them an advantage in the fermentation process (*Porto et al., 2017*). In addition, bacteria in the *Lactobacillaceae* family, including *Leuconostoc* and *Pediococcus*, occupy a significant proportion in the digestives of the small intestine and large intestine in the fermented liquid feed, indicating that the fermentation process has an important impact on the composition of intestinal microbiota (*Bunte et al., 2020*). Additionally, the presence of *Weizyella*, a genus that has been less studied, may also contribute to the functional properties of fermented products, potentially enhancing their therapeutic effects (*Chen et al., 2024*). The interplay between these microbial populations can lead to a more balanced gut microbiota, which is essential for maintaining health and preventing diseases. For instance, the modulation of gut microbiota through fermentation has been linked to improved metabolic functions and reduced inflammation, which are critical for the efficacy of traditional Chinese medicine (*Hao et al., 2024*). An increase in the abundance of *Leuconostoc* and *Pseudomonas Weissella* undoubtedly improved the ability of fermented yupingfeng san to regulate intestinal digestion and the immune response. The abundance of important pathogenic bacteria and the potential pathogen *Enterobacter* decreased significantly in yupingfeng san fermentation samples because of the low pH environment that results from lactic acid bacteria fermentation and the antibacterial activity of beneficial microorganisms and their metabolites. As the fermentation progressed, the abundance of *Pseudomonas* decreased significantly. The relative abundance of the genus was much lower in the fermented group than in the unfermented group. The decreased abundance demonstrated that fermentation is beneficial for animal health. Therefore, the observed increase in these beneficial bacteria after fermentation suggests a promising avenue for enhancing the therapeutic potential of Chinese medicinal products.

Compared with other fermented Chinese medicines, it shows richer fermentation products and potential biological functions. The fermentation process enhances the bioavailability and efficacy of these compounds, which are known for their antioxidant, anti-inflammatory, and immunomodulatory properties. For instance, studies have demonstrated that fermented products can significantly improve the antioxidant activities of herbal extracts, as seen in the fermentation of *Radix Puerariae* and red yeast rice, which resulted in higher contents of isoflavones and enhanced antioxidant potential (*Huang, Zhang & Xue, 2017*). Similarly, the fermentation of Danggui Buxue Tang with *L. plantarum* led to the conversion of flavonoid glycosides to their aglycones, which are believed to have better gut absorption and pharmacological efficacy (*Guo et al., 2020*). Furthermore, the bidirectional immunomodulatory activity of fermented polysaccharides from yupingfeng indicates that fermentation can enhance the biological activities of traditional herbal formulations, promoting their use in therapeutic applications (*Sun et al., 2017*). Overall, the fermentation of yupengfeng san not only enriches its flavonoid content but also enhances its potential health benefits, making it a promising candidate for further research and application in traditional medicine. The ability of $CD4^+T$ cells to produce IL-5 and IL-13 after IL-4 stimulation was inhibited by quercetin following activation of transcription factors NF-$\kappa$B and STAT6. The expression of cytokine mRNA was also inhibited, which
demonstrated that quercetin regulates the IL-4-mediated immune response, specifically the Th1/Th2 cytokine balance, which weakened the allergic immune response *(Tanaka et al., 2020)*. The results of an *in vitro* study showed that quercetin upregulated *IRG* and *IFNα* and downregulated *TGFβ* mRNA expression *(Ruansit & Charerntantanakul, 2020)*. Furthermore, quercetin can be used as an effective oral immunomodulator to improve the cell-mediated immune defense against highly pathogenic porcine reproductive and respiratory syndrome virus.

Multiple studies have investigated the anti-inflammatory activity of phlorin against lung and liver inflammation, and the associated antioxidant potential triggered by phlorin provides an additional mechanism to block the formation of advanced glycation end products (AGEs), thereby improving intestinal inflammatory complications *(Jeon et al., 2017; Khalifa, Bakr & Osman, 2017; Wang et al., 2018)*. The anti-inflammatory effect of phlorin has a relationship with increased antioxidant enzyme glutathione levels in HT-29 and Caco-2 colon cancer cells. It also is associated with the downregulation of IL-8 and NF-κB in DLD1 colon cancer cells. Benzoic acid is converted into 4-hydroxybenzoic acid by microorganisms after fermentation. 4-hydroxybenzoic acid has considerable antioxidant *(Cho et al., 1998)* and antibacterial characteristics, low toxicity, and a certain neuroprotective ability *(Winter et al., 2017)*. Caffeic acid also significantly increased after fermentation. A number of other studies found that caffeic acid has strong anti-inflammatory and antibacterial effects. This finding can explain the decrease in the abundance of harmful bacteria and *Bacillus* species after yupingfeng san fermentation. In addition, we also found that the content of liquiritin decreased after fermentation, whereas the content of liquiritigenin and isoglycyrrhizin increased significantly. Liquiritigenin, a metabolite of liquiritin, has similar pharmacological effects to liquiritin, but its bioavailability is higher and its effect is stronger. Isoglycyrrhizin is a flavonoid compound with anti-inflammatory, anti-oxidation, antibacterial, antiviral, immunomodulatory, and other pharmacological effects. Isoglycyrrhizin is also widely used in traditional Chinese medicine and often is used to treat hepatitis, bronchitis, allergic diseases, and other diseases *(Ramalingam et al., 2018)*. Similarly, paeoniflorin was downregulated after fermentation, whereas peonidin-3-O-glucoside was significantly upregulated. Because of its structural advantages, peonidin-3-O-glucoside has higher bioavailability, which is more conducive to its absorption and biological activity in the body *(Xiao, Muzashvili & Georgiev, 2014)*.

The TGF-β1/Smad and TLR4/NF-κB signaling pathways enable tectorigenin to inhibit pulmonary fibrosis and airway inflammation, which is promising for allergic asthma treatment *(Wang et al., 2020a; Wang et al., 2020b)*. Tectorigenin prevented gut-derived lipopolysaccharide-induced liver inflammation and restored the intestinal barrier by slowing the release of proinflammatory cytokines. By reducing hepatic lipid accumulation and inhibiting lipogenesis, tectorigenin also supported lipolysis and bile acid circulation by activating bile acid receptors and promoting bile acid synthesis. Additionally, tectorigenin can promote the growth of beneficial *Akkermansia* and inhibit harmful microbes, thus restoring high-fat-diet-induced gut microbial dysbiosis *(Duan et al., 2022)*.

Cyanidin may play an anti-inflammatory role by protecting the intestinal barrier and inhibiting the secretion of inflammatory cytokines. Therefore, cyanidin may be a potential

prophylactic or adjuvant drugs for inflammatory bowel disease. Studies have shown that *Atractyloides* III alleviated 2,4, 6-trinitrobenzene sulfonic acid-induced acute colitis by regulating oxidative stress and affecting the development of gut microbiota through the FPR1 and Nrf2 pathways. At the same time, *Atractylenolide* III inhibited the development of fibrosis in intestinal epithelial cells by activating the AMPK signaling pathway *(Ren et al., 2021)*. *Atractylodes* III reduced sepsis-induced lung injury and cell apoptosis by inhibiting FoxO1 and VNN1 proteins, and also inhibited the secretion of inflammatory factors. Atr III-H significantly improved lung function *(Fu et al., 2021)*. Cynarin was found to have a variety of pharmacological properties *(Asaad et al., 2021; Solis-Salas et al., 2021; Tang et al., 2021)*, including free radical scavenging and antioxidant, anticholinergic, reducing power, and metal binding activity *(Erikel, Yuzbasioglu & Unal, 2019)*. Cynarin also exhibited a variety of pharmacological properties against endothelial inflammation and alleviated inflammation by upregulating MKP-3 *(Kim et al., 2022)*. Rosmarinic acid has a wide range of pharmacological effects, including anti-apoptotic, antitumor, antioxidant, and anti-inflammatory effects. In this study, the content of rosmarinic acid was increased after yupingfeng san fermentation by *B. coagulans*. These anti-inflammatory effects also were verified by *in vitro* and *in vivo* studies for several inflammatory diseases, including atopic dermatitis, arthritis, and colitis *(Farhadi et al., 2023; Luo et al., 2020)*.

Genistein effectively reduced the serum proinflammatory cytokines TNF-$\alpha$ and IL-6 and inhibited the growth of methicillin-resistant *Staphylococcus aureus* *(Guo, 2023)* . Previous studies have shown its good anti-inflammatory and antibacterial properties. Scoparone has anti-inflammatory, anti-oxidation, anti-apoptosis, anti-fibrosis and lipid-lowering pharmacological properties. Given its pharmacological effects, scoparone is a potential candidate to treat liver diseases, including acute liver injury, alcoholic hepatotoxicity, fulminant hepatitis, liver fibrosis, and non-alcoholic fatty liver disease *(Hui et al., 2020a; Hui et al., 2020b)*. We detected a large number of free amino acids in fermented yupingfeng san, which demonstrated the degradation of proteins during microbial fermentation. The abundance of tryptophan, phenylalanine, histidine, methionine, valine, leucine, and isoleucine, which cannot be synthesized by the body, and their derivatives were significantly upregulated. These free amino acids are essential for the production of dopamine and adrenaline, regulatory T cells, the survival of T cells, the regulation of macrophage functions, and the maturation of B cells *(Miyajima, 2020)*. Small molecule metabolites are essential for animal intestinal health and the quality of fermented feed.

We demonstrated the internal relationship between the microbial community and metabolites in the fermentation of yupingfeng san by *B. coagulans* according to a combined analysis of 16S rDNA sequencing and metabolomics. For example, *Pediococcus* and pelargonidin-3-O-glucoside were negatively correlated, whereas quercetin was positively correlated with pelargonidin-3-O-glucoside. Currently, few related research reports have examined *Pediococcus*, and even fewer studies have studied the metabolism of active substances in herbs. Some studies have found that lactic acid bacteria generally have the ability to hydrolyze glycosidic bonds. It is speculated that *Pediococcus* may greatly increase the dissolution rate of active substances by hydrolyzing cellulose in herbs, and hydrolyzing glycosidic bonds of active substances. In addition, active substances, such as quercetin,

can be modified from quercetin-3-O-glucuronide to quercetin, which more easily exerts biological activity and improves the efficacy of traditional Chinese medicine in both dosage and effect *(Cui et al., 2016; Zheng et al., 2014)*. Furthermore, it was found that in the fermentation process, fermentation strains catabolize traditional Chinese medicine by producing a variety of enzymes, and convert many polar macromolecular compounds that cannot be directly absorbed and used by the human body into small molecular compounds. Thus, they may also produce new active substances and new functions, increase the concentration of active ingredients, and promote the dissolution of active substances *(Park et al., 2014)*. This is beneficial for the absorption and utilization of active ingredients in the human body *(Oh et al., 2015)*. For example, after fermentation, the content of liquiritin decreased, and glycyrrhizin and isoglycyrrhizin increased significantly, indicating that the active ingredients in the drug were transformed through the fermentation of probiotics, thus greatly improving the efficacy of the drug. Studies have shown that traditional Chinese medicine, *L. plantarum* (LP), and mixed fermentation of traditional Chinese medicine and LP (FTCM) can alleviate diarrhea symptoms caused by ceftriaxone sodium (CS), improve intestinal flora and barrier function, and FTCM has more advantages than traditional Chinese medicine or LP alone *(Guo et al., 2022)*. The study adopted probiotic liquid fermentation compound traditional Chinese medicine, compared with solid fermentation, the utilization rate of medicinal materials is lower, and the storage and use requirements are higher. In contrast, the solid state fermentation method used in the fermentation of yupingfeng powder by *B. coagulans* retained the endophytes of herbaceous plants during the fermentation process. The harmful bacteria were eliminated or inhibited, as the beneficial bacteria were developed and expanded. Compared with other microorganisms in nature, endophytes have good biocompatibility with host plants because of relationships developed over long-term evolution, and thus, they are not inhibited by the metabolites in the host body. Therefore, the transformation reaction can be more efficient and stable.

We believe that the fermented yupingfeng san is more likely than the unfermented yupingfeng san to improve the body's respiratory and intestinal immunity and to prevent respiratory and intestinal inflammation.

## CONCLUSIONS

In this study, 16S rDNA sequencing technology and metabolomics were used to analyze the microbial flora and metabolites of yupingfeng san at different time points during the fermentation of *B. coagulans*. The results showed that fermentation significantly changed the microbial composition and metabolic profile of yupingfeng san. Compared with the unfermented group, the relative abundance of *Cyanobacteria* decreased and *Firmicutes* increased in the fermentation group. The relative abundance of probiotics *Leuconostoc* and *Pediococcus* in fermented compound Chinese medicine was much higher than that in unfermented group. The abundance of the important pathogenic bacterium *Pseudomonas* and the potential pathogen *Enterobacter* decreased significantly in the fermentation group, indicating that fermentation is beneficial to animal health. At the same time, 315 different metabolites were identified after fermentation, the contents of olefins decreased, the

contents of acids, lipids, ketones and amino acids increased, and the contents of quercetin, paeoniflorin-3-O-glucoside, netin, iristin, anthocyanin, caffeic acid, rosmarinic acid, liquiritin, and isoliquiritinwere significantly up-regulated. It was found that *Pediococcus* was negatively correlated with Pelargonidin-3-O-glucoside and positively correlated with Quercetin. This study is helpful to elucidate the assembly mechanism and functional expression of microorganisms after Chinese traditional medicine fermentation, and provides a reference for further research on Chinese traditional medicine fermentation mechanism.

### Funding

This research was supported by the Key Research and Development Project of Henan Province of China (241111113400), the Modern Agricultural Industrial Technology System Project of Henan Province of China (HARS-22-12-G2) and the Key Veterinary Disciplines of Henan Province of China (312). The funders had no role in study design, data collection and analysis, decision to publish, or preparation of the manuscript.

### Grant Disclosures

The following grant information was disclosed by the authors:
Key Research and Development Project of Henan Province of China: 241111113400.
Modern Agricultural Industrial Technology System Project of Henan Province of China: HARS-22-12-G2.
Key Veterinary Disciplines of Henan Province of China: 312.

### Competing Interests

The authors declare there are no competing interests.

### Author Contributions

- Yu Kang conceived and designed the experiments, performed the experiments, analyzed the data, prepared figures and/or tables, and approved the final draft.
- Yanting Sun conceived and designed the experiments, performed the experiments, analyzed the data, prepared figures and/or tables, and approved the final draft.
- Jinzhong Cui conceived and designed the experiments, analyzed the data, prepared figures and/or tables, and approved the final draft.
- Yuzhen Song conceived and designed the experiments, analyzed the data, authored or reviewed drafts of the article, and approved the final draft.
- Zilong Sun analyzed the data, authored or reviewed drafts of the article, and approved the final draft.
- Huan Li analyzed the data, prepared figures and/or tables, authored or reviewed drafts of the article, and approved the final draft.
- Ruiyan Niu conceived and designed the experiments, authored or reviewed drafts of the article, and approved the final draft.

- Hongxing Qiao conceived and designed the experiments, authored or reviewed drafts of the article, and approved the final draft.

## Data Availability

The raw sequence data are available in the National Center for Biotechnology Information Sequence Read Archive (SRA): PRJNA1181122.

The metabolomics data that support the findings of this study are available at the CNGB Sequence Archive (CNSA) of China National GeneBank DataBase (CNGBdb): CNP0006469.

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
