# Peer review of "Combined microbiome and metabolomics analysis of yupingfeng san fermented by Bacillus coagulans: insights into probiotic and herbal interactions"

_PeerJ, doi:10.7717/peerj.19206_

## Round 0.1 · original submission · Major Revisions

Please revise the manuscript by following the reviewers' comments and suggestions. A point-by-point response is needed.

Reviewer 1 ·

Basic reporting

The English is clear and clear throughout the article.

Experimental design

The experimental design of this article is reasonable.

Validity of the findings

The conclusions are well stated.

Additional comments

The author analyzed the composition and metabolic profile of Yupingfengsan after the fermentation of Bacillus coagulans, which is helpful to elucidating the assembly mechanism and functional expression of microorganisms after the fermentation of traditional Chinese medicine. The whole article is relatively standardized, it is recommended to accept.

Reviewer 2 ·

Basic reporting

The title provides sufficient information but could be slightly rephrased for clarity. Consider revising to "Combined Microbiome and Metabolomics Analysis of Yupingfeng San Fermented by Bacillus coagulans: Insights into Probiotic and Herbal Interactions."

Experimental design

The introduction is concise, but additional context about Yupingfeng san's traditional use and the significance of its fermentation could improve the reader's understanding.
The role of Bacillus coagulans in enhancing the pharmacological properties of herbal medicines should be elaborated upon.

Validity of the findings

The description of experimental methods is clear, but including more specifics on the solid-state fermentation process, such as temperature, pH, and moisture content, would strengthen the reproducibility.
A brief mention of the software or tools used for 16S rDNA sequencing and metabolomics analysis would improve the methodological transparency.

Additional comments

The discussion is somewhat limited in its exploration of the implications of the findings. Expand on the potential therapeutic or functional benefits of the metabolites and microbial changes observed.
Compare the findings with previous studies on fermented herbal medicines to highlight similarities or novel insights.

Reviewer 3 ·

Basic reporting

The manuscript titled ‘Combined microbiome and metabolomics analysis of Yupingfeng San fermented by Bacillus coagulans’ utilizes 16S sequencing and metabolomics to study the microbiome and metabolites at different fermentation stages of Yupingfeng San. The topic is very interesting; however, several areas require attention before the manuscript can be accepted for publication.

Experimental design

1. The authors used 11 single figures in the manuscript. It is recommended that they combine multiple single figures into major figures based on the narrative of the study.
2. It is recommended to use a boxplot to demonstrate Shannon's H in Figure 1. Additionally, any significant differences between the groups should be indicated.
3. The authors referred to Figures 3a, 3b, and 3c in the results, but these are presented as single figures in the manuscript.
4. The grammar and format should be carefully checked, as the manuscript contains numerous errors, such as sentences that are not completed before the next begins and the absence of spaces between sentences.

Validity of the findings

1. The section on 'Data analysis: DNA extraction and 16S rDNA gene sequencing' should be rewritten with additional details, including the kits used for DNA extraction and the tools employed for 16S rDNA gene sequencing and processing. Furthermore, all tools used for 16S rDNA gene sequencing, processing, and analysis should be properly cited.
2. For beta diversity analysis (PcoA), statistical methods such as PERMANOVA or ANOSIM are required to determine whether different groups are significantly separated.

Additional comments

Minor comments:
1. In line 199, the authors used ‘the top 26 species categories’, but no microbial species were discussed there.
2. There are grammar errors in sentences in lines 301-303.

---

## Round 0.2 · Minor Revisions

Please revise your manuscript accordingly.

Reviewer 2 ·

Basic reporting

1. Abstract and Conclusion should be concise yet. But should give complete overview of the work and study.
2. Authors can use latest related works from reputed journals like IEEE/ACM Transactions, Elsevier, Inderscience, Springer, Taylor & Francis etc

Experimental design

The authors seem to disregard or neglect some important finding in results that have been achieved in paper. So, elaborate and explain the results in more details.
Improve the results and discussion section in paragraph.

Validity of the findings

In the Introduction section, the drawbacks of previous work can be discussed.
Properly define the motivation behind the work.
Improve the quality of figures.
The paper suggests some good models but lack serious comparison with existing model

Additional comments

As a conclusion, the technical content is not good. Therefore, the contribution of this article is also not satisfactory.

Reviewer 3 ·

Basic reporting

The manuscript has significantly improved after revision; however, several areas still need to be addressed before it can be accepted for publication.

Experimental design

1. The manuscript contains numerous errors, such as the absence of spaces between sentences in the paragraph on ‘Differential metabolites of Yupingfeng San pre- and post-fermentation.’ The authors must carefully review the manuscript to avoid these mistakes.
2. The ‘ggplot2’ is a package in R, not a method for drawing figures. The authors must provide specific details, such as the functions and packages in R used to create the figures or perform analyses in the Methods section.
3. The y axis in Figure 1 should be labeled with Shannon’s index.

Validity of the findings

no comment

---

## Round 0.3 · accepted · Accept

The manuscript can be accepted for publication now.

Reviewer 2 ·

Basic reporting

ok

Experimental design

ok

Validity of the findings

ok

Additional comments

ok

Reviewer 3 ·

Basic reporting

All of my comments have been addressed by the authors.

Experimental design

All of my comments have been addressed by the authors.

Validity of the findings

All of my comments have been addressed by the authors.

Additional comments

NA